# Subjective Deterioration of Physical and Psychological Health during the COVID-19 Pandemic in Taiwan: Their Association with the Adoption of Protective Behaviors and Mental Health Problems

**DOI:** 10.3390/ijerph17186827

**Published:** 2020-09-18

**Authors:** Peng-Wei Wang, Nai-Ying Ko, Yu-Ping Chang, Chia-Fen Wu, Wei-Hsin Lu, Cheng-Fang Yen

**Affiliations:** 1Department of Psychiatry, School of Medicine, College of Medicine, Kaohsiung Medical University, Kaohsiung 80708, Taiwan; wistar.huang@gmail.com (P.-W.W.); pino3015@hotmail.com (C.-F.W.); 2Department of Psychiatry, Kaohsiung Medical University Hospital, Kaohsiung 80708, Taiwan; 3Department of Nursing, College of Medicine, National Cheng Kung University, Tainan 70101, Taiwan; nyko@mail.ncku.edu.tw; 4School of Nursing, The State University of New York, University at Buffalo, New York, NY 14214-3079, USA; yc73@buffalo.edu; 5Department of Psychiatry, Ditmanson Medical Foundation Chia-Yi Christian Hospital, Chia-Yi City 60002, Taiwan

**Keywords:** physical health, psychological health, protective behaviors, mental health problems, COVID-19

## Abstract

This study aimed to determine the proportion of individuals who reported the deterioration of physical and psychological health during the coronavirus disease 2019 (COVID-19) pandemic in Taiwan. Moreover, the related factors of deterioration of physical and psychological health and the association between deterioration of health and adoption of protective behavior against COVID-19 and mental health problems were also examined. We recruited participants via a Facebook advertisement. We determined the subjective physical and psychological health states, cognitive and affective construct of health belief, perceived social support, mental health problems, adoption of protective behavior and demographic characteristics among 1954 respondents (1305 women and 649 men; mean age: 37.9 years with standard deviation 10.8 years). In total, 13.2% and 19.3% of respondents reported deteriorated physical and psychological health during the COVID-19 pandemic, respectively. Participants with higher perceived harm from COVID-19 compared with severe acute respiratory syndrome (SARS) were more likely to report the subjective deterioration of physical and psychological health, whereas respondents who were older and perceived a higher level of social support were less likely to report a deterioration of physical and psychological health. The subjective deterioration of psychological health was significantly associated with avoiding crowded places and wearing a mask. Both subjective deteriorations of physical and psychological health positively related to general anxiety.

## 1. Introduction

### 1.1. The Impact of the Coronavirus Disease 2019 Pandemic

The coronavirus disease 2019 (COVID-19) pandemic has been raging globally. As a novel respiratory infectious disease that is highly contagious, the COVID-19 pandemic has impacted physical [1] and mental health [2,3], the economy [4], education [5], quality of life [6], occupations [7], and the interpersonal relationships [8] of humans.

The first COVID-19 case in Taiwan was confirmed on 21 January 2020 [9]. Due to proactive containment and comprehensive contact tracing, the number of COVID-19 cases in Taiwan has remained lower than in other countries [10]. By 21 August 2020, Taiwan had tested 129,009 individuals. A total of 486 confirmed cases were identified, of which 55 were domestic and 7 had died [9]. Therefore, Taiwan did not impose a social lockdown. However, the pandemic has impacted the economy and unemployment rate profoundly [11]. In 2012–2013, Taiwan experienced a major outbreak of severe acute respiratory syndrome (SARS). The COVID-19 outbreak rekindled memories of SARS and caused fear among the people of Taiwan.

### 1.2. Physical and Psychological Health during the COVID-19 Pandemic

COVID-19 is a threat to the physical health of both infected individuals and the general public. A study in Canada found that 36% of the population was very or extremely concerned about the impact of COVID-19 to their health [12]. An online-based study on the general public in China found that 19% of the participants experienced physical pain or discomfort on the EuroQol-5D evaluating health-related quality of life [13]. The psychological health of the public has also been deeply affected by the COVID-19 pandemic. A review study found that both specific populations such as children, elderly, and medical personnel and the general population were harmed psychologically by imposition of strict isolation during the COVID-19 pandemic [14]. The COVID-19 pandemic might also threaten individual bodily integrity and autonomy and subsequently result in psychiatric comorbidity representing as atypical pictures, such as functional movement disorders [15]. These studies examined the cross-sectional status of physical and psychological health among people during the COVID-19 pandemic. Given that COVID-19 has impacted human lives rapidly and unprecedentedly, examining the deterioration of physical and psychological health since the pandemic began may provide insights into changes in health status during the COVID-19 pandemic.

### 1.3. Factors Related to Deteriorating Physical and Psychological Health during the COVID-19 Pandemic

Several individual and environmental factors may correlate with the physical and psychological health problems evident during the COVID-19 pandemic, such as pre-existing physical and mental health conditions [12,13,16,17,18], low income [13,18], and experiencing the profound influence of the pandemic on daily activities [13,19,20]. Determining the modifiable factors predicting the deterioration of physical and psychological health during the COVID-19 pandemic may provide evidence to develop prevention and intervention programs for the public affected by the COVID-19 pandemic.

The health belief model (HBM) can serve as a theoretical basis for determining the predictors of deteriorating physical and psychological health during the COVID-19 pandemic. The HBM proposes cognitive and affective constructs that predict whether an individual will adopt health-promoting behaviors. These include perceived susceptibility to and severity of a health problem, perceived benefits of and barriers to engaging in recommended action, and the belief in one’s ability to successfully perform a behavior [21,22]. Several studies have examined the association of cognitive and affective constructs of health beliefs with physical and psychological health during the COVID-19 pandemic. For example, perceived high vulnerability for contracting COVID-19 [23,24], perceived low survival likelihood [24], anxiety regarding contracting COVID-19 [13], and the distress caused by the uncertainty of the endpoint of the COVID-19 pandemic [25] predict physical and psychological health during the COVID-19 pandemic. However, HBM-based assessment is inadequate. Liao et al. [26] proposed cognitive and affective constructs of health beliefs concerning the risk of contracting (1) influenza A/H1N1 in 2009 and (2) respiratory infectious diseases in future epidemics or pandemics. These can be used to examine the cognitive and affective constructs of health beliefs predicting physical and psychological health during the COVID-19 pandemic.

Studies have found that levels of social support were significantly associated with self-efficacy and sleep quality and negatively associated with the degree of anxiety and stress among medical staff in China who were treating patients with COVID-19 [27]. However, the association between perceived social support and deteriorating physical health has not been well examined. Studies on the association between demographic characteristics and physical and psychological health during the COVID-19 pandemic have revealed mixed results. One study found that aging individuals had a higher risk of physical pain or discomfort and depression or anxiety [13], whereas other studies have found that young people were more likely to report mental health problems during the COVID-19 pandemic [17,18,28]. Moreover, several studies have confirmed that women are more likely to report poor mental health during the COVID-19 pandemic than men are [17,18,24,29]; however, gender difference in determining physical health during the COVID-19 pandemic has not been examined. Further study is needed to examine whether demographic factors relate to the deterioration of physical and psychological health during the COVID-19 pandemic.

### 1.4. Deterioration of Physical and Psychological Health and the Adoption of Protective Behaviors against COVID-19 and Mental Health Problems

Adopting protective behaviors, such as avoiding crowded places, washing hands frequently, and wearing a mask, are essential to prevent contracting COVID-19 and staying healthy. A two-wave study in China indicated that precautionary measures, such as maintaining hand hygiene and wearing a mask, were associated with a lower psychological impact from the outbreak and lower levels of stress, anxiety, and depression in both the initial stage of the COVID-19 outbreak [24] and four weeks later [30]. However, studies on people during the SARS epidemic have reported that respondents with a moderate level of anxiety were most likely to take comprehensive precautionary measures against the infection [31]. Moreover, the use of personal protective equipment increases the discomfort level and causes difficulties in communication [32]. There is a need of further research into the roles played by deteriorating physical and psychological health in the adoption of protective behaviors against COVID-19.

Physical symptoms and poor self-rated health status were significantly associated with a higher incidence of post-traumatic stress disorder and symptoms of stress, anxiety, and depression [30]. Both sleep problems [28] and suicidal ideation [33] are serious mental health problems in the era of COVID-19. It is reasonable to hypothesize that the deterioration of psychological health is significantly associated with sleep problems and suicidal ideation that have become more prevalent during the COVID-19 pandemic, whether the deterioration of physical health is significantly associated with sleep problems and suicidal ideation bears further exploration.

### 1.5. Aims of the Study

This study had three aims: (1) to determine the proportion of individuals who reported the deterioration of physical and psychological health during the COVID-19 pandemic in Taiwan, (2) to examine the association between cognitive and affective constructs of health beliefs and demographic characteristics and the subjective deterioration of physical and psychological health, and (3) to examine the association between subjective deterioration of physical and psychological health and adoption of protective behavior against COVID-19 and mental health problems.

## 2. Methods

### 2.1. Participants

The current investigation was based on the dataset of the Survey of Health Behaviors During the COVID-19 Pandemic in Taiwan, which was comprehensively described elsewhere [34]. Briefly, a Facebook advertisement was deployed between 10 April 2020 and 23 April 2020. We targeted the advertisement to Facebook users by location (Taiwan) and language (Chinese), where Facebook’s advertising algorithm determined which users to show our advertisement to. Facebook users who were 20 years or older and resided in Taiwan were eligible for this study. Participants reached the research questionnaire website through the Facebook advertisement, which was composed of a headline, main text, pop-up banner, and weblink. A total of 2031 respondents completed the research questionnaire; of them, 77 respondents were excluded due to missing data on any variable or being younger than 20. Data from 1954 respondents were analyzed. Figure 1 demonstrates the flowchart of study design. The Institutional Review Board (IRB) of Kaohsiung Medical University Hospital that is responsible for ethical review approved this study (KMUHIRB-EXEMPT(I) 20200011). As participation was voluntary and survey responses were anonymous, written informed consent was waived based on the approval of IRB. The participants were given no incentive for participation. We provided links to Taiwan Centers for Disease Control, Kaohsiung Medical University Hospital, and Medical College of National Cheng Kung University for participants to learn more about COVID-19 at the end of the online questionnaire. The analyses of information sources [34], sexual behaviors [35], and sleep and suicidality [36] using the dataset have been published elsewhere.

### 2.2. Measures

#### 2.2.1. Subjective Deterioration of Physical and Psychological Health during the COVID-19 Pandemic

The four-item self-perceived health questionnaire was developed by Ko et al. [37] to evaluate the physical and psychological health of the public during the SARS epidemic. For this study, the four questions were modified to evaluate the self-rated physical and psychological health of the respondent compared with those of other people before the COVID-19 outbreak and during the week before filling out the questionnaire (“How is the state of your physical/psychological health compared with other people before the COVID-19 pandemic/in the recent week?”). The questions are listed in Appendix A. The rating for each question ranged from 1 (much worse), 2 (mildly worse), 3 (the same), 4 (mildly better), and 5 (much better). Then, the self-reported physical and psychological health states were compared between before and during the COVID-19 pandemic. Respondents whose self-rated physical health score in the preceding week was lower than that before the COVID-19 outbreak were classified as having a deterioration of physical health during the COVID-19 pandemic. The respondents whose self-rated physical health score in the preceding week was the same as or higher than that before the COVID-19 outbreak were classified as having no deterioration in physical health. The respondents with or without a deterioration of psychological health during the COVID-19 pandemic were classified according to the same rules.

#### 2.2.2. Cognitive and Affective Constructs of Health Beliefs Related to COVID-19

We examined the cognitive and affective constructs of health beliefs in the context of COVID-19, according to the particularization of the HBM to respiratory infectious disease pandemics [26]. The four cognitive constructs included perceived relative susceptibility to COVID-19 (“What do you think are your chances of contracting COVID-19 over the next 1 month compared with others outside your family?”), perceived severity of COVID-19 relative to SARS (“How serious is COVID-19 relative to SARS?”), sufficiency of knowledge and information about COVID-19 (“Do you think you have sufficient knowledge and information on COVID-19?”), and perceived self-confidence in coping with COVID-19 (“How confident are you that you can cope well with COVID-19?”). The affective construct included worry about COVID-19 (“Please rate how worried you are toward COVID-19”). The questions, response scales, and dichotomous scales for statistical analysis are listed in Appendix A.

#### 2.2.3. Perceived Social Support

Three questions developed in the study of Tardy [38] were used to assess the levels of perceived social support from families, friends, and colleagues during the preceding week (“In the past 7 days, were you satisfied with the support from your (1) family, (2) friends, and (3) colleagues or classmates?). The questions and response scales are listed in Appendix A. The total score for the three questions indicates the level of perceived social support. Higher scores represent higher perceived social support. The internal reliability (Cronbach’s α) of the measure was 0.813 in this study. As the scores of perceived social support were not normally distributed (skewness = −0.138, kurtosis *=* −0.056, *p* of Kolmogorov–Smirnoff test <0.05), we used the median score of 9 as the cutoff, and respondents whose score of perceived social support was lower than 9 and whose score was 9 or higher were classified as the groups of low and high perceived social support, respectively.

#### 2.2.4. Adoption of Protective Behaviors against COVID-19

We assessed whether the participants avoided crowded places, washed their hands more often, or wore a mask more often in the preceding week to protect themselves from contracting COVID-19 (“In the past week, did you (1) avoid going to crowded places, (2) wash your hands more often, and (3) wear a mask more often?”) [26]. The questions, response scales, and dichotomous scales for statistical analysis are listed in Appendix A.

#### 2.2.5. Mental Health Problems

Respondents’ level of general anxiety was assessed with the previously validated state-anxiety scale of the Chinese version of State-Trait Anxiety Inventory (C-STAI), wherein respondents rate their feelings in response to 10 general statements (for example, “I feel rested”) [26,39,40]. A previous study found that the state-anxiety scale of C-STAI had a high internal consistency (Cronbach’s alpha = 0.90, split-half reliability = 0.89) and high item-total correlations (*r* = 0.42–0.62) [36]. Two questions adopted from the revised 5-item Brief Symptom Rating Scale were used to assess sleep problems (“In the past week, did you have sleep problems?”) and suicidal ideation (“In the past week, did you ever have suicidal thoughts?”) in the preceding week [41,42]. Previous studies confirmed that both questions had acceptable test-retest reliability (paired sample correlation coefficients = 0.73–0.78) and significant correlations with suicidal risk in general population (*p* < 0.001) [41,42]. The questions, response scales, and dichotomous scales for statistical analysis are listed in Appendix A.

#### 2.2.6. Demographic Characteristics

Data on gender (women vs. men), age, and education level (university qualifications or above vs. high school qualifications or below) were collected. As age was not normally distributed (skewness = 0.485, kurtosis *=* −0.218, *p* of Kolmogorov–Smirnoff test < 0.05), we used the median age (37 years old) as the cutoff, and respondents who were younger than 37 and who were 37 or older were classified as the younger and older groups, respectively.

### 2.3. Statistical Analysis

Data analysis was performed using SPSS 22.0 statistical software (SPSS Inc., Chicago, IL, USA). Demographic characteristics, cognitive and affective constructs of health beliefs related to COVID-19, and perceived social support were compared between respondents who did or did not exhibit a subjective deterioration in physical and psychological health during the COVID-19 pandemic using univariate logistic regression with the crude odds ratio (cOR). Furthermore, all potential predictive variables identified from the first step were eligible for inclusion in the multivariate logistic regression models with an adjusted odds ratio (aOR) to determine the independent predictors of the subjective deterioration of physical and psychological health. The association between the deterioration of physical and psychological health and adoption of protective behaviors against COVID-19 (avoiding crowded places, washing hands, and wearing a mask) and mental health problems (general anxiety, sleep problems, and suicidal ideation) was examined using multivariate logistic regression after controlling for the effects of gender, age, and educational level. Moreover, *p* values, odds ratios (ORs), and 95% confidence intervals (CIs) of OR were used to indicate significance. A two-tailed *p* value of <0.05 indicated statistical significance.

We also used the standard criteria proposed by Baron and Kenny [43] to examine whether the associations of the deteriorated physical and psychological health and related factors (cognitive and affective constructs of health beliefs, perceived social support, adoption of protective behaviors, and mental health problems) were moderated by demographic characteristics that were significantly associated with the deterioration of physical and psychological health. The interactions (demographic characteristics × related factors) were selected into the logistic regression analysis to examine the moderating effects.

## 3. Results

### 3.1. Participant Variables

Data from 1954 respondents (1305 women and 649 men) were analyzed. The mean age was 37.9 years (standard deviation [SD] = 10.8 years; range: 20–74), 1029 (52.7%) participants were classified as the older group, and 1736 (88.8%) participants had university qualifications or above. The mean scores for worry and self-confidence were 6.2 (SD = 2.2; range: 0–10) and 3.1 (SD = 0.8; range: 1–5), respectively. Regarding the cognitive and affective constructs of health beliefs related to COVID-19, 346 (17.7%) respondents reported high perceived susceptibility to COVID-19; moreover, 1379 (70.6%) perceived that COVID-19 was more harmful than SARS, 1763 (90.2%) reported having sufficient knowledge and information about COVID-19, 1686 (86.3%) reported having high confidence in coping with COVID-19, and 1228 (62.8%) reported having a high degree of worry about COVID-19. The mean level of perceived social support was 8.6 (SD = 2.0; range: 0–12), and 1189 (60.8%) participants were classified as the group of high perceived social support.

Table 1 shows the proportions of the respondents with various levels of subjective physical and psychological health and changes in health from before to during the COVID-19 pandemic. Most of the respondents reported their health the same as other people before (physical: 46.1%; psychological: 43.1%) and during the COVID-19 pandemic (physical: 55.4%; psychological: 48.2%). Regarding the changes in health from before to during the COVID-19 pandemic, 69.2% and 69.8% of the respondents reported no change in physical and psychological health, respectively. Of those who had changes in physical and psychological health, most reported mild deterioration (physical: 10.8%; psychological: 15.2%) or improvement (physical: 15.4%; psychological: 9.2%). In total, 257 (13.2%) and 377 (19.3%) respondents reported that their physical and psychological health deteriorated during the COVID-19 pandemic, respectively.

### 3.2. Factors Related to the Deterioration of Physical and Psychological Health

Table 2 presents the results of the univariate logistic regression model examining the associations between demographic characteristics, cognitive and affective constructs of health beliefs related to COVID-19, perceived social support, and the subjective deterioration of physical and psychological health. Participants who reported higher perceived harm with respect to COVID-19 than to SARS (B = 0.484, cOR = 1.623, 95% CI: 1.184–2.224, *p* = 0.003), had sufficient knowledge and information about COVID-19 (B = 0.549, cOR = 1.732, 95% CI: 1.020–2.941, *p* = 0.042), and frequently worried about COVID-19 (B = 0.349, cOR = 1.418, 95% CI: 1.068–1.882, *p* = 0.016) were more likely to report the subjective deterioration of physical health during the pandemic because all B values were larger than zero. Respondents who were older (B = −0.239, cOR = 0.746, 95% CI: 0.573–0.970, *p* = 0.029), had high confidence in coping with COVID-19 (B = −0.640, cOR = 0.527, 95% CI: 0.379–0.734, *p* < 0.001), and perceived a higher level of social support (B = −0.357, cOR = 0.700, 95% CI: 0.538–0.911, *p* = 0.008) were less likely to report a deterioration of physical health because the B values were smaller than zero. Respondents who perceived greater harm potentially resulting from COVID-19 compared with SARS (B = 0.419, cOR = 1.521, 95% CI: 1.169–1.979, *p* = 0.002) and who frequently worried about COVID-19 (B = 0.617, cOR = 1.853, 95% CI: 1.443–2.381, *p* < 0.001) were more likely to report the subjective deterioration of psychological health during the COVID-19 pandemic because the B values were larger than zero. Respondents who were men (B = −0.279, cOR = 0.757, 95% CI: 0.591–0.969, *p* = 0.027), were older (B = −0.376, cOR = 0.686, 95% CI: 0.548–0.860, *p* = 0.001), and perceived a higher level of social support (B = −0.491, cOR = 0.612, 95% CI: 0.488–0.767, *p* < 0.001), were less likely to report a deterioration of psychological health because the B values were smaller than zero.

All variables that were significantly associated with the subjective deterioration of physical and psychological health during the COVID-19 pandemic in the univariate logistic regression model were included in the multivariate logistic regression models (Table 3). The results indicate that participants with higher perceived harm from COVID-19 compared with SARS (B = 0.421, aOR = 1.524, 95% CI: 1.107–2.099, *p* = 0.010) and sufficient knowledge and information about COVID-19 (B = 0.763, aOR = 2.146, 95% CI: 1.247–3.692, *p* = 0.006) were more likely to report the subjective deterioration of physical health during the COVID-19 pandemic because both B values were larger than zero. Respondents who were older (B = −0.270, aOR = 0.763, 95% CI: 0.584–0.997, *p* = 0.048), had high confidence in coping with COVID-19 (B = −0.608, aOR = 0.545, 95% CI: 0.385–0.771, *p* = 0.001), and perceived a higher level of social support (B = −0.305, aOR = 0.737, 95% CI: 0.562–0.966, *p* = 0.027) were less likely to report a deterioration of physical health because all B values were smaller than zero. Respondents who perceived a higher degree of harm from COVID-19 compared with SARS (B = 0.339, aOR = 1.403, 95% CI: 1.073–1.835, *p* = 0.013) and more frequently worried about COVID-19 (B = 0.503, aOR = 1.653, 95% CI: 1.280–2.136, *p* < 0.001) were more likely to report a subjective deterioration of psychological health during the COVID-19 pandemic because of positive B values. Respondents who were older (B = −0.303, aOR = 0.738, 95% CI: 0.587–0.929, *p* = 0.010) and perceived a higher level of social support (B = −0.433, aOR = 0.649, 95% CI: 0.516–0.816, *p* < 0.001) were less likely to report a deterioration of psychological health because of negative B values.

The moderating effects of age on the associations between perceived harm of COVID-19 relative to SARS, sufficiency of knowledge and information about COVID-19, confidence in coping with COVID-19, and perceived social support with the deterioration of physical health were further examined based on the criteria proposed by Baron and Kenny (1986). The results demonstrate that the interaction between age and sufficiency of knowledge and information about COVID-19 was significantly associated with the deterioration of physical health (B = −1.316, aOR = 0.268, 95% CI: 0.079–0.912, *p* = 0.035), indicating that age moderated the association between the deterioration of physical health and sufficiency of knowledge and information about COVID-19. Further analysis found that the significant association between the deterioration of physical health and sufficient knowledge and information about COVID-19 existed only in younger respondents (B = 1.564, aOR = 4.776, 95% CI: 1.705–13.381, *p* = 0.003) but not in older ones (B = 0.249, aOR = 1.283, 95% CI: 0.662–2.486, *p* = 0.461).

The moderating effects of gender and age on the associations between perceived harm of COVID-19 relative to SARS, worry about COVID-19, and perceived social support with the deterioration of psychological health were also examined. The results demonstrate that the interactions between age and other factors were not significantly associated with the deterioration of psychological health, indicating that age did not moderate the associations between the deterioration of psychological health and other factors.

### 3.3. Deterioration of Health and Adoption of Protective Behaviors against COVID-19 and Mental Health Problems

Regarding the adoption of protective behaviors against COVID-19, 1587 respondents (81.2%) reported avoiding crowded places, 1511 (77.3%) washed hands more often, and 1511 (77.3%) wore a mask more often. Table 4 demonstrates the results from examining the association between the deterioration of physical and psychological health and the adoption of protective behaviors against COVID-19. The results indicate that after controlling for the effects of demographic characteristics, the subjective deterioration of psychological health was associated with more adoption of two protective behaviors, including avoiding crowded places (B = 0.411, aOR = 1.508, 95% CI: 1.088–2.092, *p* = 0.014) and wearing a mask (B = 0.525, aOR = 1.690, 95% CI: 1.238–2.308, *p* = 0.001). The interactions between demographic characteristics and the deterioration of psychological health were not significantly associated with avoiding crowded places, indicating that demographic characteristics did not moderate the associations between the deterioration of psychological health and avoiding crowded places. No significant association was found between the deterioration of physical health and adoption of protective behaviors against COVID-19.

Regarding mental health problems, 943 respondents (48.3%) had a high level of general anxiety, 1089 (55.7%) had sleep problems, and 206 (10.5%) had suicidal ideation. The results from examining the association between the deterioration of physical and psychological health and mental health problems are shown in Table 5. The results show that after controlling for the effects of demographic characteristics, the deterioration of both physical and psychological health was associated with more general anxiety (physical: B = 0.687, aOR = 1.989, 95% CI: 1.499–2.639, *p* < 0.001; psychological: B = 0.497, Aor = 1.643, 95% CI: 1.295–2.084, *p* < 0.001). The deterioration of psychological health and not physical health was associated with more sleep problems (B = 0.271, aOR = 1.312, 95% CI: 1.033–1.665, *p* = 0.026). The interactions between gender and the deterioration of physical and psychological health were not significantly associated with general anxiety. The interaction between age and the deterioration of psychological health was not significantly associated with sleep problems. The results indicate that neither gender nor age moderated the association between the deterioration of health and general anxiety and sleep problems. The deterioration of physical or psychological health was not significantly associated with suicidal ideation.

## 4. Discussion

### 4.1. Issues of Recruiting Participants from the Facebook Advertisements

Before discussing the results, some issues related to the method of recruiting participants using the Facebook advertisement warrants discussion first. Recruiting participants through Facebook can deliver large numbers of participants quickly, cheaply, and with minimal effort as compared with mail and phone recruitment [44]. Facebook is a platform that provides the opportunity to assess the general public during fast-moving infectious disease outbreaks. However, Facebook users may not be representative of the population. A review of a study that recruited respondents through Facebook reported a bias in favor of women, young adults, and people with higher education and incomes [45]. The gender disproportion of the respondents also existed in the present study. To control the effect of gender, gender was used as the covariate when we examined the associations between the deterioration of health and the adoption of protective behaviors and mental health problems. Moreover, the present study examined the moderating effects of gender. However, the nonrepresentation of the population in the study should be cautiously considered, and is a consequence of using social media to recruit the participants.

### 4.2. Deterioration of Physical and Psychological Health

This study found that 13.2% and 19.3% of respondents reported experiencing a deterioration of physical and psychological health during the COVID-19 pandemic, respectively. According to the statistics of the National Health Insurance Administration, Taiwan, the numbers of patients visiting health care facilities during the period of April to June in 2020 reduced 12.9% when compared with the same period in 2019 [46]. People with chronic illnesses may worry about contracting COVID-19 in hospitals and doctor’s offices and therefore not seek medical assistance and delay treatment. People with anxiety may interpret changes in perceived bodily sensations as symptoms of being ill, related or unrelated to COVID-19, and complain of deteriorating physical and psychological health [14]. Although Taiwan was not placed under lockdown, people may have reduced outdoor activities or stopped routine exercise due to the worry of contracting COVID-19 and the burden of physical and psychological health problems may have therefore increased [47]. The results of this study indicate that in addition to monitoring health states of people who are quarantined or have contracted COVID-19, it is necessary for the governments and health professionals to early detect health problems of and timely deliver medical assistance to the public in the pandemic. Introducing novel methods of clinical interaction, such as telemedicine and the use of electronic devices for COVID-19 education, self-assessment, and maintenance of a symptom diary may assist in overcoming the mounting challenges of the COVID-19 pandemic [20,48]. Health promotion strategies directed at adopting or maintaining positive health-related behaviors should be utilized to address the increase in psychological distress during the pandemic [18]. Moreover, promoting community-supported interventions for stress and anxiety due to COVID-19 is recommended [2].

### 4.3. Factors Related to the Deterioration of Physical and Psychological Health

This study found that the perceived harm from COVID-19, more than that from SARS, was significantly associated with the subjective deterioration of physical and psychological health during the COVID-19 pandemic. The perceived risk of contracting COVID-19 may cause stress, which may compromise physical and psychological health [49]. The public evaluates the risk of COVID-19 relative to SARS based on the information they receive from the media and social networks. This study also found that self-rated knowledge and information about COVID-19 were positively associated with the deterioration of physical health. The provision of timely and accurate information on COVID-19 is fundamental to mitigating the disease [50] and for rationally understanding COVID-19. Moreover, high confidence in coping with COVID-19 was negatively associated with deterioration of physical health. Helping build confidence to successfully cope with the pandemic by delivering information through traditional and social media should be a priority for governments and health professionals. However, controlling misinformation on COVID-19 remains a challenge.

This study found that perceived social support was negatively associated with the deterioration of physical and psychological health. Good social interactions not only provide emotional support but also daily necessities, which may contribute to the maintenance of physical and psychological health. For example, social support can increase individual capacity to maintain health behaviors [51]. A study on women’s sport practice in Spain found that brothers/sisters, best friends and workmates encourage women to practice exercise; in particular, the presence of supportive friends increases with age [52]. Social support may be attenuated due to social distancing according to the health policy requirement and the fear of contracting COVID-19. Social support can be offered through telecommunication instead of physical contact to those who have been quarantined to prevent mental health problems. The governments should take an initiative to provide support for those who were socially isolated before the pandemic.

### 4.4. Deterioration of Health and Adoption of Protective Behaviors against COVID-19 and Mental Health Problems

This study found that the respondents who reported deteriorated psychological health were more likely to avoid crowded places and wear masks. The results of previous studies were mixed. A study in Cyprus found that higher anxiety was positively associated with the adoption of measures related to personal hygiene, whereas higher depression was negatively associated with higher compliance with precautionary measures [53]. A study in China during the initial outbreak of COVID-19 demonstrated that the adoption of precautionary measures was associated with a lower psychological impact from the outbreak of COVID-19 and lower levels of stress, anxiety, and depression [24,30]. Another study in China found that people’s perceptions that the outbreak can be controlled by protective behaviors were associated with lower prevalence of depression and anxiety [54]. The results of the present and previous studies indicate that there might be factors such as the timing of survey, severity of the pandemic and definition of psychological health influencing the association between psychological health and adoption of protective behaviors.

This study found that the deterioration of both physical and psychological health was significantly associated with general anxiety and that of psychological health with sleep problems. General anxiety is closely connected to dysfunction of interoception, which can disturb the process by which the nervous system senses, interprets, and integrates signals originating from within the body, providing a moment-by-moment mapping of the body’s internal landscape across conscious and unconscious levels [55]. Somatic discomfort, such as increased muscle ache and heart rate, and psychological discomfort, such as excessive worry and irritability were also the core symptoms of generalized anxiety disorder [56]. Therefore, general anxiety and the perception of deteriorating health may occur together. Moreover, the present study found that deteriorated psychological health was significantly associated with sleep problems. Sleep disturbance is the core symptom of several mental disorders; for example, depression and anxiety disorders [56]. Sleep problems may be used as an indicator of psychological health and may warrant psychological intervention during the COVID-19 pandemic.

### 4.5. Limitations

The present study has some limitations in addition to the gender nonrepresentation of the participants recruited by the Facebook advisement. First, there might be recall bias for the health state before the COVID-19 outbreak. Second, the cross-sectional design of this study limited causal inference between changes in health state and related factors. Third, some factors such as chronic diseases that might influence deteriorated health in the COVID-19 pandemic were not examined in the present study. Fourth, the psychometric measures used in the present study for evaluating perceived social support warrants further examination.

## 5. Conclusions

This Facebook-based online study on the general public in Taiwan found that 13.2% and 19.3% of respondents reported deteriorated physical and psychological health during the COVID-19 pandemic, respectively. Both subjective deteriorations of physical and psychological health positively related to general anxiety. The results indicate that the physical and psychological health of the public, but not only those who were contracted with COVID-19, should be focus of health professionals’ concern. The present study identified several health belief constructs, social support and demographic characteristics that were significantly associated with deteriorated physical and psychological health. These factors can be used to screen for the individuals who need intervention for physical and psychological health problems. The subjective deterioration of psychological health was significantly associated with avoiding crowded places and wearing a mask. Further study is needed to examine the mechanism accounting for the association and provide reference for developing strategies to promote adoption of protective behaviors against respiratory infectious diseases.

## Figures and Tables

**Figure 1 ijerph-17-06827-f001:**
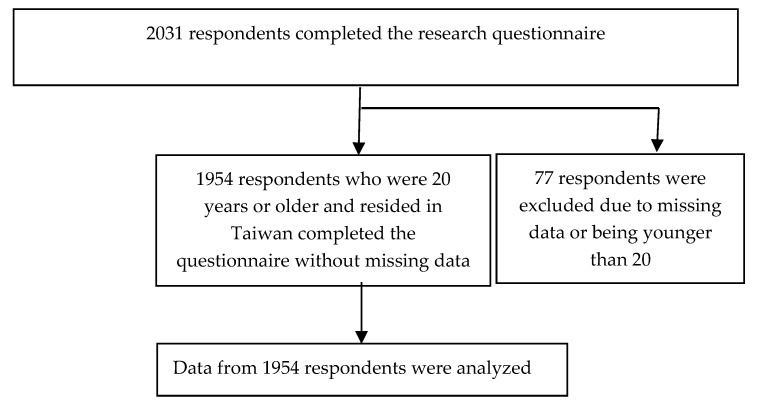
Flowchart of study design.

**Table 1 ijerph-17-06827-t001:** Subjective physical and psychological health before and during the coronavirus disease 2019 (COVID-19) pandemic (N = 1954).

	Physical Health		Psychological Health	
	before the Pandemic	during the Pandemic	before the Pandemic	during the Pandemic
	*n* (%)	*n* (%)	*n* (%)	*n* (%)
Much worse (Score 0)	52 (2.7)	36 (1.8)	62 (3.2)	73 (3.7)
Mildly worse (Score 1)	462 (23.6)	344 (17.6)	231 (11.8)	241 (12.3)
Same (Score 2)	901 (46.1)	1083 (55.4)	842 (43.1)	942 (48.2)
Mildly better (Score 3)	404 (20.7)	376 (19.2)	572 (29.3)	512 (26.2)
Much better (Score 4)	135 (6.9)	115 (5.9)	247 (12.6)	186 (9.5)
Changes from before to during the pandemic				
Deteriorated				
Severely (change for 3 scores)	8 (0.4)		8 (0.4)	
Moderately (change for 2 scores)	38 (1.9)		72 (3.7)	
Mildly (change for 1 score)	211 (10.8)		297 (15.2)	
No change	1353 (69.2)		1363 (69.8)	
Improved				
Mildly (change for 1 score)	300 (15.4)		180 (9.2)	
Moderately (change for 2 scores)	40 (2.0)		32 (1.6)	
Severely (change for 3 scores)	3 (0.2)		1 (0.1)	
Profoundly (change for 4 scores)	1 (0.1)		1 (0.1)	
Deterioration of health	257 (13.2)		377 (19.3)	
No change or improvement of health	1697 (86.8)		1577 (80.7)	

**Table 2 ijerph-17-06827-t002:** Comparisons of demographic characteristics and cognitive and affective health beliefs of COVID-19 between respondents with and without the deterioration of physical and psychological health: univariate logistic regression model (N = 1954).

	Deterioration of Physical Health						Deterioration of Psychological Health					
	No	Yes	B	cOR	95% CI	*p*	No	Yes	B	cOR	95% CI	*p*
Gender, *n* (%)												
Female (*n* = 1305)	1137 (87.1)	168 (12.9)	0.073	1.076	0.816–1.418	0.605	1035 (79.3)	270 (20.7)	−0.279	0.757	0.591–0.969	0.027
Male (*n* = 649)	560 (86.3)	89 (13.7)					542 (83.5)	107 (16.5)				
Age, *n* (%)												
Younger (*n* = 925)	787 (85.1)	138 (14.9)	−0.293	0.746	0.573–0.970	0.029	718 (77.6)	207 (22.4)	−0.376	0.686	0.548–0.860	0.001
Older (*n* = 1029)	910 (88.4)	119 (11.6)					859 (83.5)	170 (16.5)				
Education level, *n* (%)												
University or above (*n* = 1736)	1505 (86.7)	231 (13.3)	−0.125	0.882	0.573–1.359	0.570	1391 (80.1)	345 (19.9)	−0.366	0.694	0.468–1.028	0.068
High school or below (*n* = 218)	192 (88.1)	26 (11.9)					186 (85.3)	32 (14.7)				
Perceived susceptibility to COVID-19, *n* (%)												
Low (*n* = 1608)	1405 (87.4)	203 (12.6)	0.247	1.280	0.924–1.772	0.137	1308 (81.3)	300 (18.7)	0.222	1.248	0.941–1.656	0.124
High (*n* = 346)	292 (84.4)	54 (15.6)					269 (77.7)	77 (22.3)				
Perceived harm of COVID-19 relative to SARS, *n* (%)												
No (*n* = 575)	520 (90.4)	55 (9.6)	0.484	1.623	1.184–2.224	0.003	489 (85.0)	86 (15.0)	0.419	1.521	1.169–1.979	0.002
Yes (*n* = 1379)	1177 (85.4)	202 (14.6)					1088 (78.9)	291 (21.1)				
Sufficient knowledge and information about COVID-19, *n* (%)												
No (*n* = 191)	175 (91.6)	16 (8.4)	0.549	1.732	1.020–2.941	0.042	159 (83.2)	32 (16.8)	0.190	1.209	0.812–1.799	0.350
Yes (*n* = 1763)	1522 (86.3)	241 (13.7)					1418 (80.4)	345 (19.6)				
Confidence in coping with COVID-19, *n* (%)												
Low (*n* = 268)	213 (79.5)	55 (20.5)	-0.640	0.527	0.379–0.734	<0.001	207 (77.2)	61 (22.8)	−0.245	0.783	0.574–1.068	0.122
High (*n* = 1686)	1484 (88.0)	202 (12.0)					1370 (81.3)	316 (18.7)				
Worry about COVID-19, *n* (%)												
Low (*n* = 726)	648 (89.3)	78 (10.7)	0.349	1.418	1.068–1.882	0.016	627 (86.4)	99 (13.6)	0.617	1.853	1.443–2.381	<0.001
High (*n* = 1228)	1049 (85.4)	179 (14.6)					950 (77.4)	278 (22.6)				
Perceived social support, *n* (%)												
Low (*n* = 765)	645 (84.3)	120 (15.7)	−0.357	0.700	0.538–0.911	0.008	581 (75.9)	184 (24.1)	−0.491	0.612	0.488–0.767	<0.001
High (*n* = 1189)	1052 (88.5)	137 (11.5)					996 (83.8)	193 (16.2)				

CI: confidence interval; cOR: crude odds ratio; SD: standard deviation.

**Table 3 ijerph-17-06827-t003:** Factors related to the deterioration of physical and psychological health status: multivariate logistic regression model.

	Deterioration of Physical Health				Deterioration of Psychological Health			
	B	aOR	95% CI	*p*	B	aOR	95% CI	*p*
Male ^a^					−0.172	0.842	0.654–1.083	0.180
Older age ^b^	−0.270	0.763	0.584–0.997	0.048	−0.303	0.738	0.587–0.929	0.010
Perceived harm of COVID-19 more than SARS ^c^	0.421	1.524	1.107–2.099	0.010	0.339	1.403	1.073–1.835	0.013
Sufficient knowledge and information about COVID-19 ^d^	0.763	2.146	1.247–3.692	0.006				
High confidence in coping with COVID-19 ^e^	−0.608	0.545	0.385–0.771	0.001				
High worry about COVID-19 ^f^	0.183	1.201	0.896–1.611	0.220	0.503	1.653	1.280–2.136	<0.001
High social support ^g^	−0.305	0.737	0.562–0.966	0.027	−0.433	0.649	0.516–0.816	<0.001

aOR: adjusted odds ratio; CI: confidence interval; COVID-19: coronavirus disease 2019. ^a^ Women as reference; ^b^ younger age as reference; ^c^ perceived harm of COVID-19 less than SARS as reference; ^d^ not having enough knowledge and information about COVID-19 as reference; ^e^ low confidence as reference; ^f^ low worry as reference; ^g^: low social support as reference.

**Table 4 ijerph-17-06827-t004:** Association between the deterioration of physical and psychological health and adoption of protective behaviors against COVID-19.

	Avoiding Crowded Places				Washing Hands				Wearing a Mask			
	B	aOR	95% CI	*p*	B	aOR	95% CI	*p*	B	aOR	95% CI	*p*
Deterioration of physical health	0.259	1.296	0.890–1.886	0.177	0.260	1.297	0.918–1.832	0.140	0.328	1.388	0.971–1.986	0.072
Deterioration of psychological health	0.411	1.508	1.088–2.092	0.014	0.293	1.340	0.998–1.799	0.052	0.525	1.690	1.238–2.308	0.001
Male ^a^	−0.265	0.768	0.605–0.974	0.030	−0.166	0.847	0.677–1.060	0.147	−0.042	0.959	0.765–1.202	0.717
Older age ^b^	0.450	1.568	1.241–1.983	<0.001	0.390	1.477	1.188–1.837	<0.001	−0.022	0.979	0.787–1.216	0.846
Low educational level ^c^	−0.459	0.632	0.448–0.891	0.009	−0.442	0.643	0.465–0.888	0.007	−0.294	0.745	0.540–1.030	0.075

aOR: adjusted odds ratio; CI: confidence interval. ^a^ Women as reference; ^b^ younger age as reference; ^c^ high educational level as reference.

**Table 5 ijerph-17-06827-t005:** Association between the deterioration of physical and psychological health and general anxiety, sleep problems, and suicidal ideation.

	General Anxiety				Sleep Problem				Suicidal Idea			
	B	aOR	95% CI	p	B	aOR	95% CI	p	B	aOR	95% CI	*p*
Deterioration of physical health	0.687	1.989	1.499–2.639	<0.001	0.259	1.295	0.980–1.712	0.069	0.285	1.330	0.878–2.015	0.179
Deterioration of psychological health	0.497	1.643	1.295–2.084	<0.001	0.271	1.312	1.033–1.665	0.026	−0.195	0.823	0.558–1.214	0.325
Male ^a^	−0.299	0.741	0.611–0.899	0.002	−0.087	0.917	0.758–1.110	0.374	−0.002	0.998	0.728–1.369	0.990
Older age ^b^	−0.127	0.881	0.733–1.059	0.176	−0.279	0.756	0.630–0.908	0.003	−1.341	0.262	0.187–0.365	<0.001
Low educational level ^c^	0.257	1.293	0.968–1.727	0.082	0.009	1.009	0.757–1.346	0.951	0.366	1.442	0.893–2.327	0.134

aOR: adjusted odds ratio; CI: confidence interval. ^a^ Women as reference; ^b^ younger age as reference; ^c^ high educational level as reference.

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
