# Peer review of "Subjective Deterioration of Physical and Psychological Health during the COVID-19 Pandemic in Taiwan: Their Association with the Adoption of Protective Behaviors and Mental Health Problems"

_ijerph, 2020, doi:10.3390/ijerph17186827_

Round 1
Reviewer 1 Report
This contribution is an interesting work on a changing and almost unknown reality. The authors have made a great effort to provide scientific content to the situation with the COVID-19 outbreak. However, the work has to be improved to meet the IJERPH standards of excellence. In this regard, the following should be addressed:
- The Abstract has to be rewritten, trying to convey a more orderly and clear message. The first sentence is five lines long, making fluent reading difficult. In addition, basic data such as the proportions of participants according to gender and age (M and SD - as shown in lines 201-202) are missing. Another question to take into account is if the first idea to communicate is that the data collection was through the Facebook platform, this methodological question is not the main focus of the research (this same observation applies to line 122). Thus, it is recommended that the abstract follow the IMRAD format (Introduction, Methods, Results, and Discussion).
- The link on line 50 should be removed, as it does not refer to exactly the same reference 11.
- In line 103 replace “crowed” with “crowded”.
- The sentence on line 131 needs to be reworked. If the present study reproduces a previously used methodology, this is what should be said. However, if you are dealing with the same database again, it must be clearly specified how this study differs from the previous one - otherwise, the present study would not be valid.
- The information that has been placed in the supplementary material should be integrated into the text. In addition, a table should be added with the descriptive statistics of the different measures (including - depending on the case - means, standard deviations, skewness, kurtosis and the Kolmogorov-Smirnoff test).
- The description of the Results is limited to the value of p. It must also incorporate the values ​​and B and their implications in the subject studied.
In the Discussion of work results, it is not clear that “The results of this study indicate that public health, both physical and psychological, requires attention from the government and health professionals” (line 227). So, it is very important to document this affirmation. - With regards to the phrase "Good social interactions not only provide emotional support but also daily necessities, which may contribute to the maintenance of physical and psychological health", it would be interesting to cite studies that give a scientific base to this statement (as is the case of https: //www.redalyc.org/pdf/2351/235126897004.pdf). In this sense, the authors are invited to relate the explanation of their results with those of the knowledge prior to the COVID-19 outbreak.
Once these changes had been introduced, this works would increase its quality for sure.
Author Response
Reviewer 1
Comment 1
The Abstract has to be rewritten, trying to convey a more orderly and clear message. The first sentence is five lines long, making fluent reading difficult. In addition, basic data such as the proportions of participants according to gender and age (M and SD - as shown in lines 201-202) are missing. Another question to take into account is if the first idea to communicate is that the data collection was through the Facebook platform, this methodological question is not the main focus of the research (this same observation applies to line 122). Thus, it is recommended that the abstract follow the IMRAD format (Introduction, Methods, Results, and Discussion).
Response
Thank you for your comment. We rewrote the Abstract as below. Firstly, we divided the first sentence into two sentences to make reading easier. Second, we added gender and age. Third, we moved the recruitment method from the first sentence to Methods section to follow the IMRAD format. Please refer to line 20-28. We also revised the sentence in line 126 by removing the words “Facebook-based online.”
“This study aimed to determine the proportion of individuals who reported the deterioration of physical and psychological health during the coronavirus disease 2019 (COVID-19) pandemic in Taiwan. Moreover, the related factors of deterioration of physical and psychological health and the association between deterioration of health and adoption of protective behavior against COVID-19 and mental health problems were also examined. We recruited participants via a Facebook advertisement. We determined the subjective physical and psychological health states, cognitive and affective construct of health belief, perceived social support, mental health problems, adoption of protective behavior and demographic characteristics among 1954 respondents (1,305 women and 649 men; mean age: 37.9 years with standard deviation 10.8 years).”
Comment 2
The link on line 50 should be removed, as it does not refer to exactly the same reference 11.
Response
Thank you for your reminding. We removed the link. Please refer to line 51.
Comment 3
In line 103 replace “crowed” with “crowded”.
Response
Thank you for your reminding. We replace “crowed” with “crowded”. Please refer to line 107.
Comment 4
The sentence on line 131 needs to be reworked. If the present study reproduces a previously used methodology, this is what should be said. However, if you are dealing with the same database again, it must be clearly specified how this study differs from the previous one - otherwise, the present study would not be valid.
Response
Thank you for your comment. We revised the sentence as below to introduce the dataset of the Survey of Health Behaviors During the COVID-19 Pandemic in Taiwan. We also specified that the current study differs from the previous ones. Please refer to line 134-135 and line 150-151 .
“The current investigation was based on dataset of the Survey of Health Behaviors During the COVID-19 Pandemic in Taiwan, which was comprehensively described elsewhere [34].…. The analyses of information sources [34], sexual behaviors [35], and sleep and suicidality [36] using the dataset have been published elsewhere.”
Comment 5
The information that has been placed in the supplementary material should be integrated into the text.
Response
We added the questions for assessing the health belief constructs, perceived social support, and mental health problems as below into the revised manuscript.
Health belief constructs: “The four cognitive constructs included perceived relative susceptibility to COVID-19 (“What do you think are your chances of contracting COVID-19 over the next 1 month compared with others outside your family?”), perceived severity of COVID-19 relative to SARS (“How serious is COVID-19 relative to SARS?”), sufficiency of knowledge and information about COVID-19 (“Do you think you have sufficient knowledge and information on COVID-19?”), and perceived self-confidence in coping with COVID-19 (“How confident are you that you can cope well with COVID-19?”). The affective construct included worry about COVID-19 (“Please rate how worried you are toward COVID-19.”). The four cognitive constructs included perceived relative susceptibility to COVID-19 (“What do you think are your chances of contracting COVID-19 over the next 1 month compared with others outside your family?”), perceived severity of COVID-19 relative to SARS (“How serious is COVID-19 relative to SARS?”), sufficiency of knowledge and information about COVID-19 (“Do you think you have sufficient knowledge and information on COVID-19?”), and perceived self-confidence in coping with COVID-19 (“How confident are you that you can cope well with COVID-19?”). The affective construct included worry about COVID-19 (“Please rate how worried you are toward COVID-19.”).” Please refer to line 174-181.
Perceived social support: “In the past 7 days, were you satisfied with the support from your 1) family, 2) friends, and 3) colleagues or classmates?” Please refer to line 186-187.
Mental health problems: “Respondents’ level of general anxiety was assessed with the previously validated state-anxiety scale of the Chinese version of State-Trait Anxiety Inventory (C-STAI) wherein respondents’ rate their feelings in response to 10 general statements (For example, “I feel rested.”) [26, 39, 40]….Two questions adopted from the Revised 5-item Brief Symptom Rating Scale were used to assess sleep problems (“In the past week, did you have sleep problems?”) and suicidal ideation (“In the past week, did you ever have suicidal thoughts?”) in the preceding week [41, 42].” Please refer to line 202-209.
Comment 6
A table should be added with the descriptive statistics of the different measures (including - depending on the case - means, standard deviations, skewness, kurtosis and the Kolmogorov-Smirnoff test).
Response
Thank you for your comment. The numbers and rates of dichotomic variables were described in Table 1. We added the skewness, kurtosis and p values of Kolmogorov-Smirnoff test of two continuous variables (age and perceived social support) into the revised manuscript. Because that these two continuous variables were not normally distributed, we transformed them into the dichotomic variables. The significance of the new analysis results was the same as the original ones.
“Because that the scores of perceived social support were not normally distributed (skewness = -0.138, kurtosis = -0.056, p of Kolmogorov-Smirnoff test < 0.05), we used the median score of 9 as the cutoff, and respondents whose score of perceived social support was lower than 9 and whose score was 9 or higher were classified as the groups of low and high perceived social support, respectively.” Please refer to line 190-194.
“Because that age was not normally distributed (skewness = 0.485, kurtosis = -0.218, p of Kolmogorov-Smirnoff test < 0.05), we used the median age (37 years old) as the cutoff, and respondents who were younger than 37 and who were 37 or older were classified as the younger and older groups, respectively.” Please refer to line 216-219.
Comment 7
The description of the Results is limited to the value of p. It must also incorporate the values ​​and B and their implications in the subject studied.
Response
Thank you for your suggestion. We revised the contents of Results section by incorporating the values ​​and B and their implications into them. Please refer to line 271-287, line 290-305, line 331-337, and line 346-351.
Comment 8
In the Discussion of work results, it is not clear that “The results of this study indicate that public health, both physical and psychological, requires attention from the government and health professionals” (line 227). So, it is very important to document this affirmation.
Response
Thank you for your comment. We revised this sentence as below. Please refer to line 390-393.
“The results of this study indicate that in addition to monitoring health states of people who are quarantined or contracted COVID-19, it is necessary for the governments and health professionals to early detect health problems of and timely deliver medical assistance to the public in the pandemic.”
Comment 9
With regards to the phrase "Good social interactions not only provide emotional support but also daily necessities, which may contribute to the maintenance of physical and psychological health", it would be interesting to cite studies that give a scientific base to this statement (as is the case of https: //www.redalyc.org/pdf/2351/235126897004.pdf). In this sense, the authors are invited to relate the explanation of their results with those of the knowledge prior to the COVID-19 outbreak.
Response
Thank you for your comment. We added the results of previous studies prior to the COVID-19 outbreak as below to explain our results. Please refer to line 416-419.
“For example, social support can increase individual capacity to maintain health behaviors [51]. A study on women’s sport practice in Spain found that brothers/sisters, best friends and workmates encourage women to practice exercise; in particular, the presence of supportive friends increases with age [52].”
Reviewer 2 Report
COMMENTS TO AUTHORS:
This paper purposed to determine the proportion of individuals who reported the deterioration of physical and psychological health during the coronavirus disease 2019 (COVID-19) pandemic in Taiwan, the related factors of deterioration of physical and psychological health, and the association between deterioration of physical and psychological health and adoption of protective behavior against COVID-19 and mental health problems. They concluded that 13.2% and 19.3% of respondents reported deteriorated physical and psychological health during the COVID-19 pandemic, respectively. Especially, both subjective deteriorations of physical and psychological health positively related to general anxiety. I do have some comments as listed below in the order noted.
Comment 1:
The quality of the data set is very important, especially for a Facebook user. For this reason, please clarify the included criteria and excluded criteria of sample collection in the Methods section and please also provide a flowchart immediately at the subsection of Participants.
Comment2:
Please clarify and define the following outcomes: Deterioration of Physical and Psychological Health, Adoption of Protective Behaviors, and General Anxiety, Sleep Problems, and Suicidal Ideation in the subsection of Study Measures.
Author Response
Reviewer 2
Comment 1
The quality of the data set is very important, especially for a Facebook user. For this reason, please clarify the included criteria and excluded criteria of sample collection in the Methods section and please also provide a flowchart immediately at the subsection of Participants.
Response
Thank you for your suggestion. We added the included criteria and excluded criteria of sample collection as below in the Methods section. Please refer to line 136-143. We also added the flowchart of study design into the revised manuscript. Please refer to line 152-153.
“We targeted the advertisement to Facebook users by location (Taiwan) and language (Chinese), where Facebook’s advertising algorithm determined which users to show our advertisement to. Facebook users who were 20 years or older and resided in Taiwan were eligible for this study...A total of 2,031 respondents completed the research questionnaire; of them, 77 respondents were excluded due to missing data or being younger than 20. Data from 1,954 respondents were analyzed.”
Comment 2
Please clarify and define the following outcomes: Deterioration of Physical and Psychological Health, Adoption of Protective Behaviors, and General Anxiety, Sleep Problems, and Suicidal Ideation in the subsection of Study Measures.
Response
Thank you for your comment. In revised manuscript we added the questions for assessing the deterioration of physical and psychological health, adoption of protective behaviors, general anxiety, sleep problems, and suicidal ideation into the revised manuscript.
Deterioration of physical and psychological health: “How is the state of your physical/psychological health compared with other people before the COVID-19 pandemic/in the recent week?” Please refer to line 161-162.
Adoption of protective behaviors: “In the past week, did you 1) avoid going to crowded places, 2) wash your hands more often, and 3) wear a mask more often?” Please refer to line 198-199.
General anxiety, sleep problems, and suicidal ideation: “Respondents’ level of general anxiety was assessed with the previously validated state-anxiety scale of the Chinese version of State-Trait Anxiety Inventory (C-STAI) wherein respondents’ rate their feelings in response to 10 general statements (For example, “I feel rested.”) [26, 39, 40]….Two questions adopted from the Revised 5-item Brief Symptom Rating Scale were used to assess sleep problems (“In the past week, did you have sleep problems?”) and suicidal ideation (“In the past week, did you ever have suicidal thoughts?”) in the preceding week [41, 42].” Please refer to line 202-209.
Reviewer 3 Report
Thank you for the interesting article.
I think that the topic is relevant and appropriate for the journal.
I saw that the general construction of the article is good.
The major concern in accepting the article in the present version is about the sample at the study and the chosen web-tool. Consequently, the conclusion and discussion reported are extremely biased by the characteristics of the responders.
In general, I think that the tools for the collection of data in the field of medicine need to be chosen with extreme caution. People are frequently submitted to market surveys and it is, therefore, difficult for one individual to distinguish between a medical or non-medical inquiry.
In particular, by Facebook, you selected a highly imbalanced sample in favor of the female gender (the double!). This element is very questionable and deserves a more defined criticism that needs to be outlined in the results, discussion, conclusion, and limits sections.
Another element that arouses my perplexity is the interpretation of the statistically significant difference of age between the groups. I wonder how you could consider a 38,1-year-old mean group and a 36,0 yr-old mean group as being so different in age as to represent different generations? They are commonly considered as nearly equal in age; otherwise please, explain to me what is the life experience in your country that allows you to define that group as ‘older’ less deteriorated in physical and psychological contexts.
The third observation is about a limit that you reported regarding the lack of information about the sample’s health state (a very important limit!!).
Did you not think to join a brief questionnaire about the major categories of diseases in your survey? In this way, you could have compared subjects affected or not by some diseases.
In consideration of the importance of the topic in this historical context and the necessity of having the major range of data possible, I think that your paper can be published BUT after rewriting the discussion and consequently the conclusions and outlining the limits more clearly.
Through the text, below are my detailed observations.
Introduction
Page 2 line 56-62: join also that besides the effects on general people’s physical and psychological health, the COVID-19 is leading to the increased onset of psychiatric events (see Piscitelli, D., Perin, C., Tremolizzo, L., Peroni, F., Cerri, C., & Cornaggia, C. (2020). Functional movement disorders in a patient with COVID-19. NEUROLOGICAL SCIENCES).
MEASURES:
page 4 line 148: “ responses were compared with those of others……” please clarify what this sentence means. Others who?
page 4 line 151: “ ….score was lower….. than before....” that is: was a score lower than 1 point or more enough? It is questionable to define ‘lower ‘a score with a delta of 1 in a questionnaire. ….”.score in the preceding week”….Did the subjects fill the questionnaire two times?
RESULTS:
page 5 line 201: please put in the range of age, not only the mean
“ line 203: worry and self-confidence= 6,1 precise the level of this score (score low, high
etc..)
“ line 210: ….”.8,6”…... same as above
“ line 224: “...were older” see the comments in my preamble, (slightly older?)
page ?6? instead of 1 (there is a mistake in the enumeration of the pages) line 240 ….’older’... see above
“ line 249-251: “.....avoiding……...protective behavior against….” This is not clear to me: what do you intend to say? Are not ‘avoiding crowing etc.’ protective behaviors?
DISCUSSION
At first here I suggest to remarque that the sample has a high sex ratio imbalance
page ?7?.instead of 1 (there is a mistake in the enumeration of the pages)..
line 269: here I suggest to compare your percentage with other studies otherwise the data is
not informative.
Line 271-272: you do not have any element to state this sentence because you did not collect data about sample diseases.
Through the text, when you talk about the age, please explain why the two groups can answer in a different way regarding, for example, the social support and comment why the two groups can experience such a different perception.
page ?8? line 324-329: I do not see the correspondence of these sentences in the results.
LIMITS:
line 336: if Facebook is a promising tool for surveys on health problems, please justify why you have a so great bias in the sample, as sex is.
line 340-341: please, outline vigorously this limit!!
Author Response
Reviewer 3
Comment 1
The major concern in accepting the article in the present version is about the sample at the study and the chosen web-tool. Consequently, the conclusion and discussion reported are extremely biased by the characteristics of the responders. In general, I think that the tools for the collection of data in the field of medicine need to be chosen with extreme caution. People are frequently submitted to market surveys and it is, therefore, difficult for one individual to distinguish between a medical or non-medical inquiry. In particular, by Facebook, you selected a highly imbalanced sample in favor of the female gender (the double!). This element is very questionable and deserves a more defined criticism that needs to be outlined in the results, discussion, conclusion, and limits sections.
Response
Thank you for your reminding. To address the issue of gender disproportion, the present study made the efforts below.
- To controlling the effect of gender, gender was used as the covariate when we examined the associations between the deterioration of health and the adoption of protective behaviors and mental health problems. Please refer to line 232.
“The association between the deterioration of physical and psychological health and adoption of protective behaviors against COVID-19...and mental health problems...was examined using multivariate logistic regression after controlling for the effects of gender, age, and educational level.”
- We added the examination for the moderating effects of gender on the associations of the deteriorated physical and psychological health and their related factors (cognitive and affective constructs of health beliefs and perceived social support), adoption of protective behaviors, and mental health problems in the revised manuscript. We added the related paragraph in Methods and Results in the revised manuscript as below.
Methods:
“We also used the standard criteria proposed by Baron and Kenny (1986) [43] to examine whether the associations of the deteriorated physical and psychological health and related factors (cognitive and affective constructs of health beliefs, perceived social support, adoption of protective behaviors, and mental health problems) were moderated by demographic characteristics that were significantly associated with the deterioration of physical and psychological health. The interactions (demographic characteristics x related factors) were selected into the logistic regression analysis to examine the moderating effects.” Please refer to line 235-241.
Results
“The moderating effects of age on the associations between perceived harm of COVID-19 relative to SARS, sufficiency of knowledge and information about COVID-19, confidence in coping with COVID-19, and perceived social support with the deterioration of physical health were further examined based on the criteria proposed by Baron and Kenny (1986). The results demonstrated that the interaction between age and sufficiency of knowledge and information about COVID-19 was significantly associated with the deterioration of physical health (B = -1.316, aOR = 0.268, 95% CI: 0.079-0.912, p = 0.035), indicating that age moderated the association between the deterioration of physical health and sufficiency of knowledge and information about COVID-19. Further analysis found that the significant association between the deterioration of physical health and sufficient knowledge and information about COVID-19 existed only in younger respondents (B = 1.564, aOR = 4.776, 95% CI: 1.705-13.381, p = 0.003) but not in older ones (B = 0.249, aOR = 1.283, 95% CI: 0.662-2.486, p = 0.461).” Please refer to line 306-317.
“The moderating effects of gender and age on the associations between perceived harm of COVID-19 relative to SARS, worry about COVID-19, and perceived social support with the deterioration of psychological health were also examined. The results demonstrated that the interactions between age and other factors were not significantly associated with the deterioration of psychological health, indicating that age did not moderate the associations between the deterioration of psychological health and other factors.” Please refer to line 318-323.
“The interactions between gender and the deterioration of physical and psychological health were not significantly associated with general anxiety. The interaction between age and the deterioration of psychological health was not significantly associated with sleep problem. The results indicated that neither gender nor age moderated the association between the deterioration of health and general anxiety and sleep problem.” Please refer to line 351-356.
- In the beginning of Discussion section the revised manuscript, we added a new paragraph discussing this important issue. Please refer to line 365-378.
Discussion
“4.1. Issues of Recruiting Participants from the Facebook Advertisements
Before discussing the results, some issues related to the method of recruiting participants using the Facebook advertisement warrants discussion first. Recruiting participants through Facebook can deliver large numbers of participants quickly, cheaply, and with minimal effort as compared with mail and phone recruitment [43]. Especially, Facebook is a platform that provides the opportunity to assess the general public during fast-moving infectious disease outbreaks. However, Facebook users may not be representative of the population. A review of a study that recruited respondents through Facebook reported a bias in favor of women, young adults, and people with higher education and incomes [44]. The gender disproportion of the respondents also existed in the present study. To controlling the effect of gender, gender was used as the covariate when we examined the associations between the deterioration of health and the adoption of protective behaviors and mental health problems. Moreover, the present study examined the moderating effects of gender. However, the nonrepresentative of the population should be cautiously considered in the study using social media to recruit the participants.”
Comment 2
Another element that arouses my perplexity is the interpretation of the statistically significant difference of age between the groups. I wonder how you could consider a 38,1-year-old mean group and a 36,0 yr-old mean group as being so different in age as to represent different generations? They are commonly considered as nearly equal in age; otherwise please, explain to me what is the life experience in your country that allows you to define that group as ‘older’ less deteriorated in physical and psychological contexts.
Response
Thank you for your comment. We transformed age into a dichotomic variable by using the median (37 years old) as the cutoff (younger than 37 vs. 37 or older). We reanalyzed the data treating age as the dichotomic variable in the revised manuscript. The significance of the new analysis results was the same as the original ones.
“… we used the median age (37 years old) as the cutoff, and respondents who were younger than 37 and who were 37 or older were classified as the younger and older groups, respectively.” Please refer to line 217-219.
Comment 3
The third observation is about a limit that you reported regarding the lack of information about the sample’s health state (a very important limit!!). Did you not think to join a brief questionnaire about the major categories of diseases in your survey? In this way, you could have compared subjects affected or not by some diseases.
Response
The present study did not examine the respondents’ health state before the COVID-19 pandemic. We listed it as one of the limitations of this study. Please refer to line 454-456.
“Third, some factors such as chronic diseases that might influence deteriorated health in the COVID-19 pandemic were not examined in the present study.”
Comment 4
In consideration of the importance of the topic in this historical context and the necessity of having the major range of data possible, I think that your paper can be published BUT after rewriting the discussion and consequently the conclusions and outlining the limits more clearly.
Response
Thank you for your suggestion. We rewrote the contents of Methods, Results and Discussion (including Limitations) described above. We also revised other contents of Discussion section based on your Comments 15-21 listed below.
Comment 5
Introduction
Page 2 line 56-62: join also that besides the effects on general people’s physical and psychological health, the COVID-19 is leading to the increased onset of psychiatric events (see Piscitelli, D., Perin, C., Tremolizzo, L., Peroni, F., Cerri, C., & Cornaggia, C. (2020). Functional movement disorders in a patient with COVID-19. NEUROLOGICAL SCIENCES).
Response
We added the reference into the revised manuscript. Please refer to line 62-64.
“The COVID-19 pandemic might also threaten the individuals’ bodily integrity and autonomy and subsequently result in psychiatric comorbidity representing as atypical pictures such as functional movement disorders [15].”
Comment 6
MEASURES:
page 4 line 148: “responses were compared with those of others……” please clarify what this sentence means. Others who?
Response
We added the questions into the revised manuscript to clarify this sentence as below. Please refer to line 158-162.
“For this study, the four questions were modified to evaluate the self-rated physical and psychological health of the respondent compared with those of other people before the COVID-19 outbreak and in the preceding week (“How is the state of your physical/psychological health compared with other people before the COVID-19 pandemic/in the recent week?”).”
Comment 7
page 4 line 151: “….score was lower….. than before....” that is: was a score lower than 1 point or more enough? It is questionable to define ‘lower ‘a score with a delta of 1 in a questionnaire. ….”.score in the preceding week”….Did the subjects fill the questionnaire two times?
Response
- In the revised manuscript, we added a new table to introduce the changes in physical and psychological health from before to during the COVID-19 pandemic in detail, and we add description for the results as below. Most of the respondents reported no change in health; of those who reported changes in physical and psychological health, most reported mild deterioration or improvement. Please refer to line 255-264.
“Table 1 shows the proportions of the respondents with various levels of subjective physical and psychological health and changes of health from before to during the COVID-19 pandemic. Most of the respondents reported their health the same as other people before (physical: 46.1%; psychological: 55.4%) and during the COVID-19 pandemic (physical: 43.1%; psychological: 48.2%). Regarding the changes of health from before to during the COVID-19 pandemic, 69.2% and 69.8% of the respondents reported no change in physical and psychological health, respectively. Of those who had changes in physical and psychological health, most reported mild deterioration (physical: 10.8%; psychological: 15.2%) or improvement (physical: 15.4%; psychological: 9.2%). In total, 257 (13.2%) and 377 (19.3%) respondents reported that their physical and psychological health deteriorated during the COVID-19 pandemic, respectively.”
- The respondents filled the questionnaire once. There might be recall bias for the health state before the COVID-19 outbreak. We have listed it as one of the limitations of this study. Please refer to line 453-454.
“Second, there might be recall bias for the health state before the COVID-19 outbreak.”
Comment 8
RESULTS:
line 201: please put in the range of age, not only the mean
Response
We put the range of age (range: 20-74) into the revised manuscript. Please refer to line 245.
Comment 9
line 203: worry and self-confidence= 6,1 precise the level of this score (score low, high etc..)
Response
We put the ranges of worry (range: 0-10) and self-confidence (range: 1-5) into the revised manuscript. Please refer to line 247. We also listed the numbers of the respondents with various levels of worry and self-confidence as below in the manuscript. Please refer to line 251-252.
“…1,686 (86.3%) reported having high confidence in coping with COVID-19, and 1,228 (62.8%) reported having a high degree of worry about COVID-19.”
Comment 10
line 210: ….”.8.6”…... same as above
Response
We put the range of perceived social support (range: 0-12) into the revised manuscript. Please refer to line 253.
Comment 11
line 224: “...were older” see the comments in my preamble, (slightly older?)
Response
We transformed age into a dichotomic variable and reanalyzed the data. Please refer to the response to Comment 2.
Comment 12
page ?6? instead of 1 (there is a mistake in the enumeration of the pages)
Response
Thank you for your reminding. We deleted the enumeration of the pages because of errors made by the automatic patching system of the Journal. Instead, the line numbers were used to represent the location of revision.
Comment 13
line 240 ….’older’... see above
Response
We transformed age into a dichotomic variable and reanalyzed the data. Please refer to the response to Comment 2.
Comment 14
line 249-251: “.....avoiding……...protective behavior against….” This is not clear to me: what do you intend to say? Are not ‘avoiding crowing etc.’ protective behaviors?
Response
Thank you for your comment. We revised this sentence as below. Please refer to line 331-333.
“…the subjective deterioration of psychological health was associated with more adoption of two protective behaviors, including avoiding crowded places (B = 0.411, aOR = 1.508, 95% CI: 1.088-2.092, p = 0.014) and wearing a mask (B = 0.525, aOR = 1.690, 95% CI: 1.238-2.308, p = 0.001).”
Comment 15
DISCUSSION
At first here I suggest to remarque that the sample has a high sex ratio imbalance
Response
In the beginning of Discussion section the revised manuscript, we added a new paragraph discussing this important issue. Please refer to the response to Comment 1.3.
Comment 16
page ?7?.instead of 1 (there is a mistake in the enumeration of the pages).
Response
Thank you for your reminding. We deleted the enumeration of the pages because of errors made by the automatic patching system of the Journal. Instead, the line numbers were used to represent the location of revision.
Comment 17
line 269: here I suggest to compare your percentage with other studies otherwise the data is not informative.
Response
To the best of our knowledge, this study is the only one to examine the changes in subjective physical and psychological health during the COVID-19 pandemic. We hope there will be other studies examining this issue, and then we can make the comparison.
Comment 18
Line 271-272: you do not have any element to state this sentence because you did not collect data about sample diseases.
Response
We added the data of the National Health Insurance Administration, Taiwan as below into the revised manuscript. Please refer to line 381-385.
“According to the statistics of the National Health Insurance Administration, Taiwan, the numbers of patients visiting health care facilities during the period of April to June in 2020 reduced 12.9% compared with the same period in 2019 [46]. People with chronic illnesses may worry about contracting COVID-19 in hospitals and doctor’s offices and therefore not seek medical assistance and delay treatment.”
Comment 19
Through the text, when you talk about the age, please explain why the two groups can answer in a different way regarding, for example, the social support and comment why the two groups can experience such a different perception.
Response
Thank you for your comment. We transformed age and perceived social support into dichotomic variables by using the median as the cutoff.
“… we used the median age (37 years old) as the cutoff, and respondents who were younger than 37 and who were 37 or older were classified as the younger and older groups, respectively.” Please refer to line 217-219.
“…we used the median score of 9 as the cutoff, and respondents whose score of perceived social support was lower than 9 and whose score was 9 or higher were classified as the groups of low and high perceived social support, respectively.” Please refer to line 192-194.
Comment 20
page ?8?
Response
Thank you for your reminding. We deleted the enumeration of the pages because of errors made by the automatic patching system of the Journal. Instead, the line numbers were used to represent the location of revision.
Comment 21
line 324-329: I do not see the correspondence of these sentences in the results.
Response
Thank you for your comment. We revised the paragraph and added the references into the revised manuscript as below. Please refer to line 440-448.
“General anxiety is closely connected to dysfunction of interoception, which can disturb the nervous system senses, interprets, and integrates signals originating from within the body, providing a moment-by-moment mapping of the body's internal landscape across conscious and unconscious levels [55]. Somatic discomfort, such as increased muscle ache and heart rate, and psychological discomfort, such as excessive worry and irritability were also the core symptoms of generalized anxiety disorder [56]. Therefore, general anxiety and the perception of deteriorating health may occur together. Moreover, the present study found that deteriorated psychological health was significantly associated with sleep problems. Sleep disturbance is the core symptom of several mental disorders; for example, depression and anxiety disorders [56].”
Comment 22
LIMITS:
line 336: if Facebook is a promising tool for surveys on health problems, please justify why you have a so great bias in the sample, as sex is.
Response
In the beginning of Discussion section the revised manuscript, we added a new paragraph discussing this important issue. Please refer to the response to Comment 1.3.
Comment 23
line 340-341: please, outline vigorously this limit!!
Response
In the beginning of Discussion section the revised manuscript, we added a new paragraph discussing this important issue. Please refer to the response to Comment 1.3.
Reviewer 4 Report
Comments for author:
This is a very interesting study presenting data on people in Taiwan and their subjective experience of their own physical and mental health. Some improvements are suggested below.
Introduction
- “A study in China found that 19% of the participants experienced physical pain or discomfort” A study of whom, and physical pain in relation to what? (L56)
- Please correct minor spelling and grammar errors – e.g., “Adopting protective behaviors, such as avoiding crowed places”. (L103)
- The line at the end of each of the paragraphs in the introduction g., “xxxx. warrants further study” is repetitive. Either rephrase, or leave this sentence off and summarise them all at the end of the introduction.
Methods:
- “This study was approved by the Institutional 136 Review Board of Kaohsiung Medical University Hospital (KMUHIRB-EXEMPT(I) 20200011). Please clarify if this is an ethical review board?
- What is meant by not requiring “informed consent” – this seems unusual and requires further explanation as to what is meant by it, and why it was not required. Are you saying you did not explain the study to participants, or people did not have to consent to agree to participate? Please be explicit in what is meant.
- The measures and their selection need further explanation – are these pre-existing measures of anxiety, sleep problems, and suicidal ideation? If so, please name the scales and provide psychometric data from the previous studies to demonstrate they are valid and reliable. If they were not, explanation as to why existing measures were not chosen. In addition, provide the psychometric data for the social support scale.
- Supplementary table S1: Does the measure of physical/psychological health ask respondents to compare themselves with others AND themselves before the pandemic began, but in the previous week? The translation is a little tricky, and confusing – I wonder if this can be clarified to improve understanding?
Results
- Perhaps “patient variables” should be “participant variables” ?
- “77 of the original 2,031 participants were excluded due to missing data.” Missing data on what? All variables, or just some. Please briefly describe.
Discussion
- No comments here.
Conclusion
- Perhaps include the context of the study (i.e., The present study of community members in Taiwan recruited over Facebook found that….)
Author Response
Reviewer 4
Introduction
Comment 1
“A study in China found that 19% of the participants experienced physical pain or discomfort” A study of whom, and physical pain in relation to what? (L56)
Response
Thank you for your comment. We revised the sentence as below. Please refer to line 57-59.
“An online-based study on the general public in China found that 19% of the participants experienced physical pain or discomfort on the EuroQol-5D evaluating health-related quality of life [13].”
Comment 2
Please correct minor spelling and grammar errors – e.g., “Adopting protective behaviors, such as avoiding crowed places”. (L103)
Response
We corrected it into “crowded” in the revised manuscript. Please refer to line 107. We also checked thorough the manuscript to correct spelling and grammar errors.
Comment 3
The line at the end of each of the paragraphs in the introduction g., “xxxx. warrants further study” is repetitive. Either rephrase, or leave this sentence off and summarise them all at the end of the introduction.
Response
Thank you for your comment. We rephrased these sentences as below.
- “However, the association between perceived social support and deteriorating physical health has not been well examined.” Please refer to line 94-95.
- “Further study is needed to examine whether demographic factors relate to the deterioration of physical and psychological health during the COVID-19 pandemic.” Please refer to line 103-104.
- “There is a need of further research into the roles played by deteriorating physical and psychological health in the adoption of protective behaviors against COVID-19.” Please refer to line 115-117.
- “…whether the deterioration of physical health is significantly associated with sleep problems and suicidal ideation bears further exploration.” Please refer to line 123-124.
Methods:
Comment 4
“This study was approved by the Institutional 136 Review Board of Kaohsiung Medical University Hospital (KMUHIRB-EXEMPT(I) 20200011). Please clarify if this is an ethical review board?
Response
Yes, this is an ethical review board. We revised this sentence as below to clarify it. Please refer to line 143-145.
“The Institutional Review Board (IRB) of Kaohsiung Medical University Hospital that is responsible for ethical review approved this study (KMUHIRB-EXEMPT(I) 20200011).”
Comment 5
What is meant by not requiring “informed consent” – this seems unusual and requires further explanation as to what is meant by it, and why it was not required. Are you saying you did not explain the study to participants, or people did not have to consent to agree to participate? Please be explicit in what is meant.
Response
Thank you for your reminding. We revised this sentence as below to clarify the meaning. Please refer to line 145-146.
“Because participation was voluntary and survey responses were anonymous, written informed consent was waived based on the approval of IRB.”
Comment 6
The measures and their selection need further explanation – are these pre-existing measures of anxiety, sleep problems, and suicidal ideation? If so, please name the scales and provide psychometric data from the previous studies to demonstrate they are valid and reliable. If they were not, explanation as to why existing measures were not chosen. In addition, provide the psychometric data for the social support scale.
Response
- We added explanations for the psychometric data of the measures for anxiety, sleep problems, and suicidal ideation as below. Please refer to line 202-212.
“Respondents’ level of general anxiety was assessed with the previously validated state-anxiety scale of the Chinese version of State-Trait Anxiety Inventory (C-STAI) wherein respondents’ rate their feelings in response to 10 general statements (For example, “I feel rested.”) [26, 39, 40]. A previous study found that the state-anxiety scale of C-STAI had a high internal consistency (Cronbach's alpha = 0.90, split-half reliability = 0.89) and high item-total correlations (r = 0.42-0.62) [36]. Two questions adopted from the Revised 5-item Brief Symptom Rating Scale were used to assess sleep problems (“In the past week, did you have sleep problems?”) and suicidal ideation (“In the past week, did you ever have suicidal thoughts?”) in the preceding week [41, 42]. Previous studies confirmed that both questions had acceptable test-retest reliability (paired sample correlation coefficients = 0.73-0.78) and significant correlations with suicidal risk in general population (p < 0.001) [41, 42].”
- Although we examined the internal reliability (Cronbach’s α = 0.813) of the measure assessing perceived social support in this study, the psychometric of the measure warrants further examination. We listed it as one of the limitations in the present study as below. Please refer to line 456-457.
“Fourth, the psychometric of the measure used in the present study for evaluating perceived social support warrants further examination.”
Comment 7
Supplementary table S1: Does the measure of physical/psychological health ask respondents to compare themselves with others AND themselves before the pandemic began, but in the previous week? The translation is a little tricky, and confusing – I wonder if this can be clarified to improve understanding?
Response
Thank you for your comment. In the revised manuscript we added the introduction for how we assessed and compared the respondents’ physical and psychological health before the COVID-19 pandemic and during the week before filling out the questionnaire as below. Please refer to line 158-165.
“For this study, the four questions were modified to evaluate the self-rated physical and psychological health of the respondent compared with those of other people before the COVID-19 outbreak and during the week before filling out the questionnaire (“How is the state of your physical/psychological health compared with other people before the COVID-19 pandemic/in the recent week?”). …Then the self-reported physical and psychological health states were compared between before and during the COVID-19 pandemic.”
Results
Comment 8
Perhaps “patient variables” should be “participant variables” ?
Response
We corrected it into “participant variables.” Please refer to line 243.
Comment 9
“77 of the original 2,031 participants were excluded due to missing data.” Missing data on what? All variables, or just some. Please briefly describe.
Response
We revised this sentence as below. Please refer to line 142-143.
“77 respondents were excluded due to missing data on any variable or being younger than 20.”
Comment 10
Conclusion
Perhaps include the context of the study (i.e., The present study of community members in Taiwan recruited over Facebook found that….)
Response
Thank you for your suggestion. We revised the first sentence of Conclusion section as below. Please refer to line 459.
“This Facebook-based online study on the general public in Taiwan found…”
Round 2
Reviewer 1 Report
It is proposed that the manuscript be published in its most recent version. The authors have made an exhaustive review making possible to better visualize the importance of the results obtained.